# Assessing the environmental factors affecting the sustainability of Aini Falaj system

**Khalifa M. Al-Kindi**  *

UNESCO Chair of Aflaj Studies, Arco-hydrology, University of Nizwa, Nizwa, Oman

* alkindi.k@unizwa.edu.om

**Data Availability Statement:** All relevant data are within the manuscript and its Supporting Information files.

**Funding:** This project is funded by the Sultanate of Oman's Ministry of Higher Education Innovation and Research (BFP/RGP/EBR/22/010).

## Abstract

This study investigates the spatial distribution patterns and environmental factors influencing the Aini Falaj system in a specific study area. The research findings are presented through the lens of the following four categories: collinearity diagnostics, spatial autocorrelation analysis, kernel density (KD) findings, and multivariate geographically weighted regression (MGWR) analysis. The collinearity diagnostics were applied to examine the interrelationships among 18 independent environmental variables. The results indicate the absence of significant multicollinearity concerns, with most variables showing values below the critical threshold of five for variance inflation factors (VIFs). The selected variables indicate minimal intercorrelation, suggesting that researchers should be confident utilizing them in subsequent modelling or regression analyses. A spatial autocorrelation analysis using Moran's Index revealed positive spatial autocorrelation and significant clustering patterns in the distribution of live and non-functional Aini Falajs. High concentrations of live or dead Falajs tended to be surrounded by neighbouring areas with similar characteristics. These findings provide insights into the ecological preferences and habitat associations of Aini Falajs, thereby aiding conservation strategies and targeted studies. The kernel density (KD) analysis depicted distribution patterns of live and dry Aini Falajs through hotspots and cold spots. Specific regions exhibited high-density areas of live Falajs, indicating favourable environmental conditions or historical factors contributing to their concentrated distribution. Identifying these high-density zones can enhance our understanding of the spatial patterns and potential factors influencing the prevalence and sustainability of Aini Falajs. The multivariate geographically weighted regression (MGWR) models revealed strong associations between the live or dead status of Aini Falajs and environmental factors. The precipitation, topographic wetness index (TWI), aspect and slope exerted positive impacts on the live status, while evaporation, solar radiation, distance to drains and drain density exerted negative influences. Similar associations were observed for the dead status, emphasising the importance of controlling evaporation, shading mechanisms, proper drainage planning and sustainable land-use practices. This study provides valuable insights into the spatial distributions and factors influencing the live and dead status of Aini Falajs, thereby contributing to our understanding of their ecological dynamics and guiding conservation efforts and management strategies.

**Competing interests:** The authors have declared that no competing interests exist.

## Introduction

The Aini Falaj system (AFS) is an ancient method for extracting and distributing groundwater in arid regions. It involves the excavation of underground tunnels and channels to transport water from natural springs to agricultural fields and communities [1]. This system enables sustainable irrigation and supports agriculture in water-scarce areas. The AFS not only supplies water to crops, but it also provides other advantages. As water travels through the channels, it naturally filters out pollutants and silt, increasing soil fertility and overall agricultural land quality. In addition, the AFS encourages water conservation [2]. By using gravity to transport water, it minimises the need for energy-intensive pumping and mitigates both costs and environmental impacts. It is a sustainable solution that ensures the efficient utilisation of water resources [3].

Despite the presence of modern irrigation techniques, farmers in Oman have persisted in utilising the AFS due to its cost-effectiveness and sustainability [4]. However, this traditional irrigation system sometimes encounters challenges that could potentially compromise its long-term viability and reliability [5]. One significant concern is the declining groundwater levels. The increasing demand for water due to population growth and economic development has led to a diminishing water table, which could potentially result in a reduced water supply for Aini Falajs [6]. This might have a severe influence on agricultural output and local residents' capacity to sustain their way of live. Due to changes in use of land, this system has challenges in Oman [7]. Demand for land rises as the country urbanizes, potentially causing changes in land use patterns. As a result, the available land area suited for Aini Falaj irrigation may be reduced, thus limiting water supply for agricultural reasons. Contamination of water sources is also a big issue for Aini Falaj. Pollution poses serious health hazards to farmers and consumers, and it has the potential to degrade the quality of crops cultivated under the AFS [8]. Furthermore, the AFS faces maintenance and repair issues. Regular maintenance is required for proper operation; however, a shortage of competent labor and finance impedes maintenance efforts. Moreover, the impact of climate change has exacerbated these concerns. Altered rainfall patterns and extreme weather events affect water availability and can cause damage to the infrastructure of Aini Falaj. Over time, these factors can diminish the effectiveness and reliability of the system [9].

Despite the considerable body of research concerning the threats confronting the aflaj systems in Oman [10, 11], the current understanding primarily relies on isolated case studies. This limited approach hinders an accurate and comprehensive assessment of the challenges faced by these traditional irrigation systems. Furthermore, there is a noticeable imbalance in the literature, with some regions receiving extensive scholarly attention while others remain woefully overlooked. The inherent variability in the data collection methods and research objectives employed across different studies has contributed to an incomplete dataset that has failed to capture the intricate and dynamic nature of the AFS in the country. Hence, there is a critical need for comprehensive investigations to unravel the intricate biological factors that impact the overall well-being of the AFS. Gaining a thorough understanding requires rigorous research that accounts for the geographic distribution, historical context and present conditions of these systems, along with their life cycle dynamics. Notably, prior research has mostly focused on qualitative and descriptive analyses, with just a few studies utilizing advanced spatial approaches for data analysis and modeling, such as geographic information systems, remote sensing, and spatial machine learning. As a result, further investigations employing these sophisticated approaches are required to expand our understanding of this system.

There are various advantages to investigating the spatial distribution of aflaj systems in Oman using geographical information systems (GIS), remote sensing, and machine learning

methodologies. These innovative methodologies allow researchers to acquire a full understanding of the distribution, structure, and environmental interactions of irrigation systems such as aflaj [12]. GIS has the ability to make spatial data easier to handle, analyze, and display, making it easier to detect patterns and relationships within the Aini system and its surrounding environment. Remote sensing provides valuable information on land cover, vegetation health, and water resources, aiding in overall system health and recognizing potential problems. Furthermore, machine learning approaches may be used to evaluate vast datasets, uncover hidden patterns, and make accurate predictions, therefore expanding our understanding of the Aini system's dynamics and responses to environmental changes. The use of these approaches yields important insights and fosters evidence-based decision-making for the long-term management and preservation of Oman's Aini system [13].

The research aims can be divided into two groups. First, sophisticated geospatial approaches were used to predict the spatial distribution, interactions, and correlations between the Aini system and the surrounding environmental factors that have a substantial influence on it. The purpose of this technique was to gain insights into the system's complicated linkages. Second, the study aimed to use these modeling insights to precisely anticipate the geographical distributions of the Aini system inside the specified study region. The overarching goal of the research was to enhance our understanding of the intricate Aini system and to develop a predictive model that would have practical significance in sustainable land use planning and management. By leveraging GIS, remote sensing and machine learning, this research contributes to advancing our knowledge of the Aini system and provides a valuable tool for decision-makers in sustainable land use planning and management.

## Study area

The northern part of Oman showcases the remarkable Al Hajar Mountains, a prominent mountain range that dominates the landscape. Amongst its peaks, Jebel Shams is the highest summit in Oman, reaching an elevation of approximately 3,009 meters above sea level [14]. These majestic mountains not only shape the topography but also significantly influence the local climate, creating a distinct environment characterised by cooler temperatures and higher precipitation compared to the surrounding regions. Within the northern region, deep wadis, which are fertile valleys formed by seasonal water flow, carve through the rugged terrain. These wadis serve as vital water sources for agricultural activities and support diverse ecosystems, providing a stark contrast to the arid landscapes found in other parts of the country. The varied topography of the northern region offers a playground for outdoor enthusiasts and ample opportunities for hiking, rock climbing and the exploration of unique geological formations.

Due to its higher elevation and proximity to the coast, the northern part of Oman experiences a more temperate climate compared to the arid desert regions. Summers in the area are generally warm, with temperatures ranging from 25˚C to 35˚C, while winters bring cooler temperatures ranging from 10˚C to 20˚C. The elevated altitude contributes to increased cloud cover and precipitation, particularly during the winter months, thus enhancing the distinct climate of the region (Fig 1).

## Materials and methods

### Aini Falaj data

The Ministry of Agriculture, Fisheries Wealth and Water Resources assumed the role of the authoritative repository for the Aini Falaj system within the designated study area. This dataset encompasses exhaustive information pertaining to the geospatial positioning and population

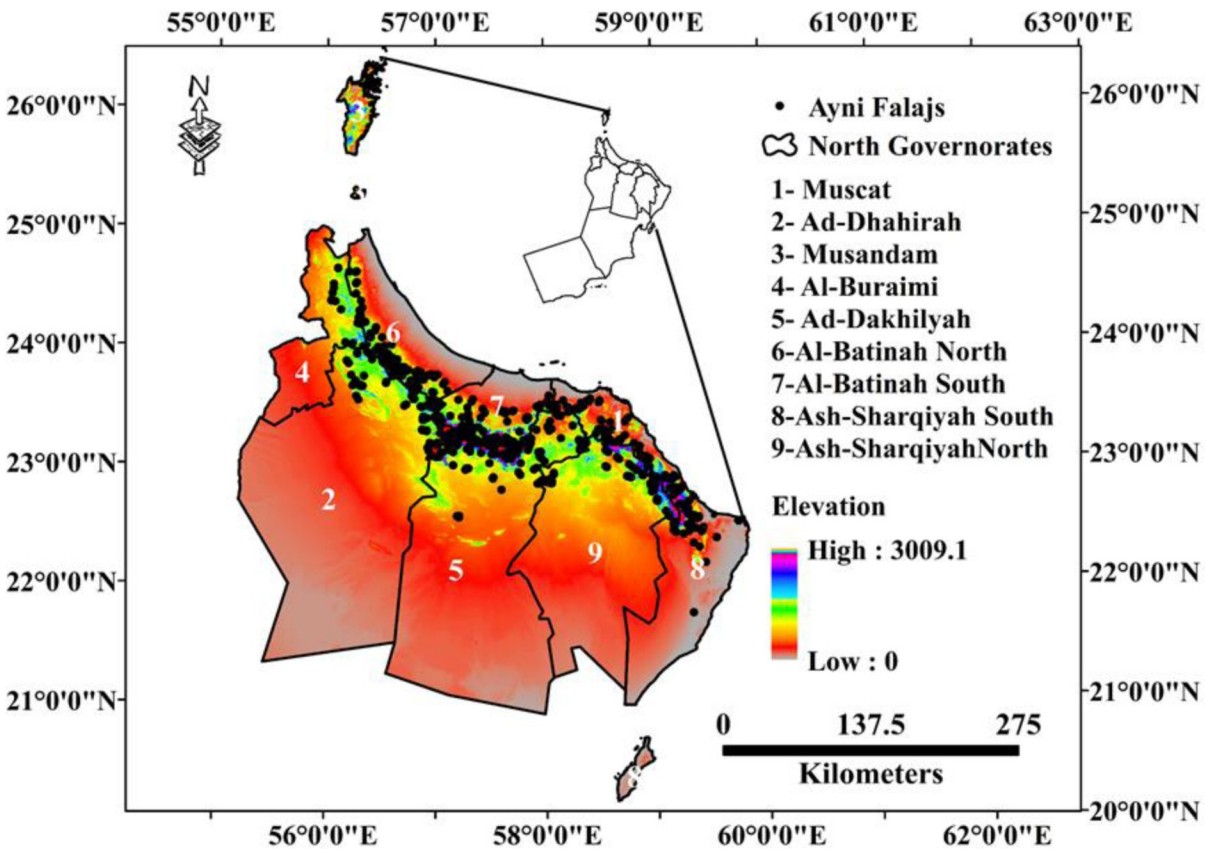

**Fig 1. Study area including the spatial distribution of the Aini Falajs and elevation feature (Esri ArcGIS Pro 2023).**

density of the Aini system, which can be conveniently accessed via the online platform hosted at the ministry's website (https://www.maf.gov.om). An initial inventory of Aini Falaj distributions was conducted in September 2001, serving as a preliminary examination of their prevalence and inherent characteristics. To guarantee the veracity and contemporaneity of the dataset, a subsequent update was executed in March 2023, thereby affording a comprehensive comprehension of the prevailing state of the system within the study area. To effectuate the conversion of the inventory data from its original Excel spreadsheet format into a more geographically informed GIS format, specifically the shapefile specification, the computational software ArcGIS Pro 3.0 (https://pro.arcgis.com/) was conscientiously employed.

## Environmental variables data

Initially, a digital elevation model (DEM) with a resolution of 30 metres was employed to extract a set of crucial variables (https://earthexplorer.usgs.gov/). These variables included the slope, aspect, slope aspect, topographic wetness index (TWI), stream power index (SPI), topographic position index (TPI), drainage density index, distance to drainage density index, plan curvature and profile curvature. The extraction process involved the utilisation of spatial analysis tools and functions available within ArcGIS Pro 3.0. In addition to the aforementioned variables, historical climate data spanning from 1970 to 2020 was acquired from WorldClim (https://www.worldclim.org/). This dataset included pertinent climate factors, such as solar radiation, evaporation and precipitation [14]. Data regarding soil type and agricultural soil were sourced from the Ministry of Agriculture, Fisheries, and Water Resources (https://www.

maf.gov.om), while geological data were obtained from Petroleum Development Oman (PDO) (https://www.pdo.co.om/). Sentinel-2 satellite data was also utilised to analyse the vegetation cover in the study area (https://scihub.copernicus.eu/) [15]. By integrating these datasets, a comprehensive understanding of the environmental factors influencing the geographic distribution of the AFS was attained, enabling the assessment of its performance and sustainability. All datasets and GIS layers were projected onto WGS 1984 UTM zone 40 (https://maps.omniscale.com/).

## Slope

The slope of the land is a critical determinant impacting the performance and sustainability of groundwater [16]. It plays a pivotal role in governing the direction and gradient of water flow, which is vital for efficient water distribution within the fields. Steep slopes pose the risk of erosion, while gentle slopes can result in waterlogging and soil salinisation. Furthermore, the slope characteristics of the surrounding landscape significantly influence the recharge of aquifers, consequently affecting the long-term sustainability of the aflaj systems [17]. Therefore, a comprehensive understanding and meticulous consideration of slope parameters are paramount in the design and management of the Aini system to ensure optimal water flow and conservation. Fig 2A visually illustrates the slope distribution observed within the study area.

## Aspect

Aspect is a crucial topographic factor impacting the performance and sustainability of aflaj systems [18]. It refers to the direction a slope faces, which influences the solar radiation, microclimate and water availability. North-facing slopes are cooler and wetter, while south-facing slopes are warmer and drier. Aspect also affects the timing and distribution of precipitation, thus impacting aquifer recharge [19]. Considering aspect alongside other topographic factors is vital for optimising Aini system design and management, as it ensures sustainable agriculture practices. Fig 2B illustrates the distribution of aspect in the study area.

## Slope aspect

The slope aspect factor, referring to the compass direction a slope faces, can impact groundwater availability [20]. North-facing slopes typically retain more moisture, which enhances groundwater recharge, while south-facing slopes may experience higher evaporation rates, potentially reducing groundwater availability. Slope aspect also affects runoff patterns, vegetation growth, evapotranspiration, and soil moisture retention. Fig 2C illustrates the spatial distributions of the slope aspect factor in the study area.

## Topographic wetness index (TWI)

The TWI is pivotal for comprehending and effectively managing groundwater resources [21]. It serves as a terrain-based metric to estimate the saturation degree and soil moisture content within a given area by analysing the land slope, aspect, and elevation. It is relevant to the AFS because of its direct impact on water flow, infiltration, and recharge rates within this system. By leveraging the TWI, researchers can identify regions with elevated moisture content, predict water accumulation potential and assess the risk of soil erosion in the AFS. This valuable information enables the optimised placement and design of AFS while facilitating sustainable land use planning and management by identifying both risks and opportunities. The TWI is a vital tool for comprehending the hydrological and ecological processes shaping Aini Falajs and

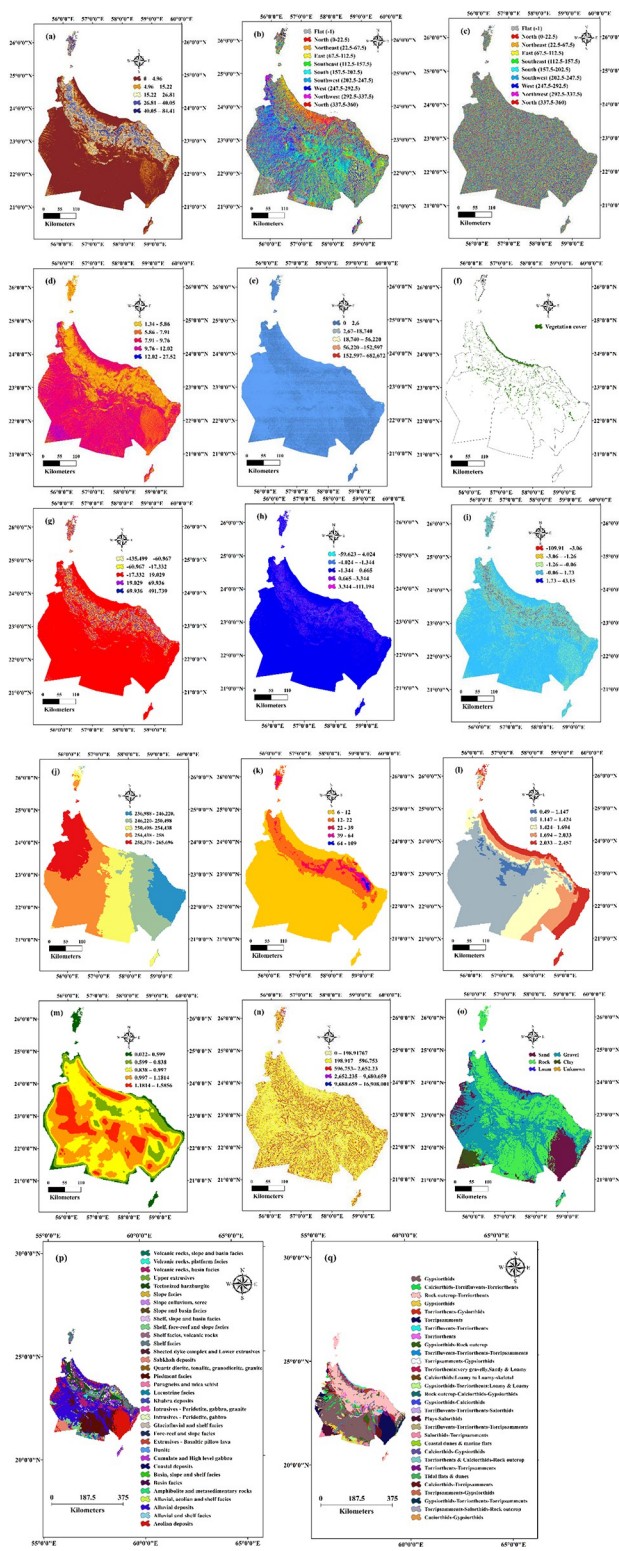

**Fig 2.** Environmental variable data including a) slope, b) aspect, c) slope aspect, d) TWI, e) SPI, f) CV, g) TPI, h) profile curvature, i) plan curvature, j) solar radiation, k) precipitation, l) evaporation, m) drainage density, n) distance to drainages, o) soil map, p) geological map and q) agricultural soil (Esri ArcGIS Pro 2023).

their surrounding environments. Fig 2D provides a visual representation of the spatial distribution of the AFS in the study area.

## Stream power index (SPI)

Fig 2E illustrates the stream power index (SPI) distribution within the study area, which serves as a significant metric for comprehending the characteristics of the AFS. SPI quantifies the erosive potential of a stream or river and is influenced by factors such as slope, drainage area, soil hydraulic properties and vegetation cover [22]. A higher SPI value suggests a greater proclivity for erosion and sediment transport, with the Aini Falaj System being heavily influenced by these environmental variables. Excessive sediment transport can result in the blockage of channels and diminished flow capacity within the underground channels of the Aini system, consequently restricting water availability for irrigation and other purposes [23]. Thus, comprehending the SPI values of surrounding streams and rivers or dry rivers (wadis) becomes crucial in the effective design and management of the AFS. By considering the SPI in conjunction with other topographic and hydrological factors, it becomes possible to optimise the flow capacity and sediment transport within the AFS, thereby promoting sustainable agricultural practices.

## Vegetation cover (VC)

The presence and characteristics of vegetation cover (VC) significantly impact the functionality and sustainability of aflaj systems [24]. VC refers to the abundance and composition of plant life in the vicinity of AFS. Vegetation plays a crucial role in regulating the hydrological cycle and enhancing water availability within the system [25]. It intercepts rainfall, reduces surface runoff and facilitates infiltration, thereby replenishing the aquifer and ensuring a consistent water flow in the aflaj systems. Furthermore, vegetation exerts control over microclimatic conditions, including temperature and humidity, which directly influence crop growth and productivity in the aflaj systems. Additionally, vegetation cover provides habitat and food resources for diverse animal species, contributing to vital ecological processes such as pollination and pest control within the aflaj systems. Therefore, a comprehensive understanding of vegetation cover and its intricate relationship with the AFS is imperative for the effective design, management and promotion of sustainable agriculture, while conserving the area's biodiversity. For this study, Sentinal-2 data with a resolution of 10 metres were utilised to calculate the VC across the study area (https://scihub.copernicus.eu/). Fig 2F presents the spatial distribution of VC within the study area.

## Topographic position index

The Topographic Position Index (TPI) is a key instrument for monitoring groundwater availability, notably within the framework of the AFS [26]. TPI provides useful insights into groundwater recharge and accumulation potential by analyzing the relative position of a site depending on elevation [27]. It assists in identifying suitable areas for groundwater extraction and plays a crucial role in promoting sustainable water management practices. Fig 2G illustrates the spatial distribution of TPI in the study area.

## Profile curvature

Profile curvature is important in evaluating groundwater viability [28], including the Aini system. It refers to the rate of slope change along the contour lines of the land surface, influencing the flow of groundwater and its movement within the subsurface. By analysing the profile

curvature, areas of increased or decreased groundwater flow can be identified. High positive curvature indicates convergence and potential groundwater accumulation [29], making them suitable for water extraction through falajs. Conversely, areas with negative curvature indicate divergence and potential groundwater recharge zones, which replenish underground aquifers. Analysing profile curvature can help to identify favourable locations for groundwater extraction, plan the distribution of AFS and ensure the sustainable utilisation of groundwater resources. Fig 2H illustrates the profile curvature in the study area.

## Plan curvature

Plan curvature, a factor in the design of aflaj systems [30], influences water flow direction and irrigation distribution. It guides the alignment and routing of falaj channels. In areas with convex plan curvature, falaj channels can follow contour lines for natural water flow, thus minimising erosion and ensuring efficient distribution. In concave areas, channels may cut across contour lines for proper water delivery. Optimising falaj channel alignment with plan curvature promotes efficient flow, prevents erosion and ensures uniform irrigation. Considerations such as slope, soil type and crop requirements complement plan curvature in Aini design and promote water efficiency and agricultural productivity [31]. Fig 2I displays the plan curvature distribution in the study area.

## Solar radiation

Solar radiation plays a crucial role in the functioning and efficiency of groundwater [32]. The availability and intensity of solar radiation directly influence the evaporation rates, water temperature and overall water balance within the aflaj systems. Solar radiation provides the energy needed for evaporation, which contributes to the cooling and transport of water through the falaj channels. The intensity and duration of solar radiation determine the evaporation rates, thus affecting the amount of water lost from the system. Understanding solar radiation patterns helps in managing water resources and optimising irrigation schedules within the AFS. Moreover, solar radiation influences water temperature, which affects the growth and development of crops in the falaj fields. Different crops have varying requirements for solar radiation, and proper solar exposure is essential for their photosynthesis and productivity. Monitoring solar radiation enables farmers to make informed decisions regarding crop selection, planting schedules and shading practices to optimise crop growth within the AFS. Fig 2J illustrates the solar radiation distribution in the study area.

## Precipitation

Precipitation plays a crucial role in the effectiveness of AFS as an irrigation system [33]. The Aini system's functionality relies on a consistent and reliable water supply, primarily from precipitation. The precipitation quantity and distribution directly impact water availability. Higher precipitation regions support the falaj system, while lower precipitation areas may require supplementary measures. Efficient water management and irrigation techniques optimise resources. AFS design considers local precipitation patterns to ensure adequate water for crops. Precipitation patterns influence irrigation scheduling. Fig 2K depicts the spatial distribution of precipitation within the study area.

## Evaporations

Evaporation significantly impacts aflaj systems and reduces the water available for irrigation and efficiency. High evaporation rates lead to increased water loss from soil and canals,

requiring additional water inputs. Evaporation is influenced by temperature, humidity, wind speed and solar radiation. Regions with high temperature, low humidity and strong winds experience higher evaporation rates, which affects aflaj systems [34]. To mitigate evaporation, water conservation practices such as mulching should be employed, and irrigation should be scheduled during cooler times. Efficient methods such as drip irrigation or sprinkler systems minimise surface evaporation. Reservoirs or storage ponds should be constructed to reduce evaporation losses. Accounting for evaporation and implementing water management strategies can help optimise AFS efficiency, minimise loss and ensure sufficient water supply. Evaporation should be considered alongside precipitation and transpiration for a sustainable AFS. Fig 2L illustrates average evaporations from 1970 to 2020.

## Drainages density

Drainage density and the Aini system are not inherently interconnected, as drainage density pertains to the density of natural dry river (wadi) or stream networks, whereas the AFS represents a man-made irrigation system [35]. However, the proximity and presence of rivers and streams can influence the design considerations of the AFS. Areas characterised by higher drainage density offer potential water sources that can be leveraged for the development of the AFS [36]. Incorporating the natural drainage patterns and the accessibility of water sources plays a crucial role in strategically designing falaj channels to efficiently distribute water throughout agricultural fields. Although drainage density itself does not exert a direct impact on the functioning of the AFS, it exerts an indirect influence by providing viable water sources for the system's establishment. The precise relationship between drainage density and the Aini Falaj is contingent upon factors such as local topography, water source availability and the intricate design of the falaj network. To visually comprehend the spatial distribution of drainage density within the study area, refer to Fig 2M. This figure offers a graphical representation that aids in understanding the varying patterns of drainage density and its potential implications for the Aini system's development.

## Distance to drainages

The proximity of aflaj systems to drainages is important for efficient irrigation. Placing aflaj systems near rivers or streams minimises water transport distance and improves water availability [37]. Shorter falaj channels reduce water loss and enhance distribution efficiency. Being situated close to drainages ensures a consistent and reliable water source, thus supporting sustainable agriculture. Accounting for the distance to drainages is crucial in Aini design to optimise water transport, efficiency and reliability. Fig 2N displays the distance to drainages in the study area.

## Soil types

Soil type plays a significant role in the performance and management of aflaj systems [38]. It directly affects water movement, infiltration rates and water-holding capacity within the underground channels. The choice of soil type in constructing and maintaining aflaj systems is crucial for efficient water distribution and for preventing issues such as clogging and water loss. Different soil types have varying permeability and water retention capabilities, which impact the overall functionality of the system. Understanding soil types along the falaj path helps identify potential challenges and design appropriate solutions, enabling optimised water distribution and sustainable agricultural practices [39]. Fig 2O presents the spatial distribution of soil types within the study area.

## Geological map

Fig 2P shows the geological map in the study area. The geological composition of an area has a significant impact on the hydrogeological characteristics and water availability [40]. Geological maps provide valuable information about the distribution of rock types, formations, and structures, which influence groundwater storage, flow patterns, and aquifer recharge. By studying geological maps, hydrogeologists and engineers can identify potential water sources, such as aquifers or groundwater reservoirs, that contribute to the sustainability of AFS. Understanding the geological features helps determine suitable locations for capturing and diverting water, optimizing the overall design and alignment of the falaj system [41].

## Agricultural soil

Agricultural soils play a crucial role in supporting plant growth by providing nutrients, water retention, and a conducive environment for root development [42]. The choice of agricultural soil can be directly impacted the efficiency of water distribution within the aflaj systems. Selecting suitable soils along the Falaj path is crucial for optimal water infiltration and avoiding issues such as waterlogging or inadequate moisture levels. Different soil types have varying capacities for water retention, drainage, and nutrient availability, which can affect crop health and productivity. Considering soil characteristics during Aini design and construction is essential. Assessing soil texture, fertility, and water-holding capacity helps in selecting appropriate soil types and implementing effective management practices. Soil amendments and proper soil management techniques enhance fertility, optimize water distribution, prevent soil erosion, and improve crop productivity [43]. Integrating knowledge of agricultural soils and the Aini system empowers farmers to make informed decisions on crop selection, irrigation practices, and soil conservation, promoting sustainable agriculture and efficient water utilization for the long-term viability of the traditional irrigation system. Fig 2Q shows the spatial distribution of the agricultural soil in the study area.

## Collinearity diagnostics

Collinearity diagnostics (CD), also known as multicollinearity diagnostics, is a statistical technique used to identify and evaluate multicollinearity among predictor variables in a regression model. Methods such as examining the correlation matrix, calculating the Variance Inflation Factor (VIF), and assessing eigenvalues and the condition number of the predictor variable matrix are commonly employed [44]. The objective is to address multicollinearity effectively, ensuring the reliability and accuracy of the statistical analysis [45].

## Spatial machine learning methods

### Spatial autocorrelation analysis

To study the geographical patterns of AFS, spatial autocorrelation analysis was used. This research allows for the examination and quantification of the degree to which the distribution of Aini Falajs exhibits geographical clustering or dispersion [46]. It is feasible to get insights into the underlying processes and variables driving the geographical organization of the Falajs by evaluating the spatial autocorrelation of their positions. The findings of this study provide to a better understanding of the spatial dynamics of Aini Falajs and their distribution patterns in the studied region.

## Kernel density

Kernel Density (KD) analysis was used to determine the intensity of AFS in the studied region. This spatial statistical approach can be used to produce a quantitative assessment of the density or concentration of falajs throughout a geographic region [47]. The generated KD surface showed regions with greater or lower densities, providing information about the geographical distribution and strength of the falajs. Following that, the KD analysis findings were coupled with environmental factors to ease spatial connection approaches.

## Multiscale geographically weighted regression

To examine the impact of environmental variables on the geographical distribution of AFS in the research region, the Multiscale Geographically Weighted Regression (MGWR) approach was utilized. By splitting the research region into smaller parts and constructing regression models for each unit, MGWR enables for the investigation of localized connections [48]. Slope, aspect, SPI, TWI, TPI, solar radiation, evaporation, plan curvature, profile curvature, precipitation, temperature, soil characteristics, agricultural soil, geological maps, distance to drainages, drainage density, and vegetation cover were all considered predictors of AFS performance. MGWR incorporates multiple spatial scales in its equation, which takes the form:

$$Y(i) = \beta 0(j)(i) + \beta 1(j)(i)X1(i) + \beta 2(j)(i)X2(i) + \ldots + \beta p(j)(i)Xp(i) + \varepsilon(i) \qquad (1)$$

The coefficients β0(j)(i) to βp(j)(i) vary across locations and scales. Estimating these coefficients involves an iterative process of solving weighted least squares problems at each location and scale. MGWR allows for flexible modeling of spatial relationships, capturing variations at different geographic levels [49].

## Results

### Collinearity diagnostics

The multicollinearity analysis investigated the interrelationships among eighteen independent environmental variables influencing Aini Falaj in the study area. The Variance Inflation Factors (VIFs) were examined to assess the severity of multicollinearity. Notably, the VIF values for the majority of variables remained below the critical threshold of 5, signifying the absence of significant multicollinearity concerns. The variables evaporation, solar radiation, and precipitation displayed moderate levels of correlation with VIF values of 2.08, 2.09, and 2.23, respectively. Similarly, VC, distance to drainages, drainages density, soil types, geological factors, and SPI exhibited VIF values ranging from 1.07 to 1.38, indicating minimal intercorrelation. It is worth noting that the variable representing agricultural soil demonstrated a VIF of 1.07, indicating its negligible contribution to multicollinearity (Table 1).

Furthermore, variables associated with topographic characteristics, including slope aspect, aspect, slope, TWI, plan curvature, profile curvature, and TSI, displayed VIF values ranging from 1.1 to 2.34, indicating acceptable levels of collinearity (Table 1). Thus, the comprehensive multicollinearity analysis demonstrated that the selected independent environmental variables, including agricultural soil, exhibited VIF values below the critical threshold. These findings underscore the absence of significant multicollinearity concerns and affirm the unique contribution of each variable in the analysis. As a result, these variables can be utilized with confidence in subsequent modeling or regression analyses, as the detrimental effects of excessive intercorrelation are minimal.

**Table 1. The multicollinearity analysis investigated the interrelationships among eighteen independent environmental variables influencing Aini Falaj in the study area.**

| Variables | VIF |
|---|---|
| Evaporation | 2.08 |
| Solar radiation | 2.09 |
| Precipitation | 2.23 |
| VC | 1.17 |
| Distance to drainages | 1.14 |
| Drainage density | 1.38 |
| TWI | 2.34 |
| SPI | 1.35 |
| Slope Aspect | 1.1 |
| Aspect | 1.1 |
| Slope | 2.27 |
| Soil types | 1.13 |
| Plan curvature | 1.72 |
| Profile curvature | 1.57 |
| Geological | 1.18 |
| TSI | 1.3 |
| Agricultural soil | 1.07 |

## Spatial autocorrelation analysis findings

The Global Moran's I Summary was conducted to assess the spatial distribution patterns of live (active) Aini and dry (non-functional) Aini Falajs. The Moran's Index values, along with other statistical measures, were employed to examine the degree of spatial autocorrelation and clustering within each category. The Moran's Index value for the live Aini Falajs category was 0.218, showing positive spatial autocorrelation and the existence of considerable clustering in the distribution of live Aini Falajs across the research region (Z-score = 16.43, p 0.0001). This finding implies that regions with significant densities of live Aini Falajs are likely to be surrounded by areas with comparable characteristics. The low variance of 0.00017 indicates that this category has a consistent and stable clustering structure (Fig 2A). The Moran's Index value for non-functional Aini Falajs was much higher at 0.708, showing larger positive spatial autocorrelation and a more severe clustering pattern when compared to the live Aini Falajs group (Z-score = 53.09, p 0.0001). This finding implies that areas with a high concentration of dead Aini Falajs are more likely to be surrounded by neighbouring areas with similar attributes. The low variance of 0.00017 indicates a consistent and stable clustering pattern within the distribution of dead Aini Falajs (Fig 2B).

## KD findings

Fig 3 illustrates KD maps, visually representing the distribution patterns of Aini Falajs within the study area. The maps effectively identify hotspots and cold-spots, denoting regions with high and low densities of live and dry Aini Falajs, respectively (Fig 4A and 4B). Hotspots indicate concentrated clusters where Aini Falajs are densely populated, exhibiting higher density in relation to surrounding areas. These hotspots offer insights into areas of intensified aggregation and significant population abundance of Aini Falajs. On the other hand, cold spots represent regions with lower densities, indicating relative scarcity or infrequent occurrence of Aini Falajs. The utilization of KD maps provides crucial information on spatial patterns and concentrations of Aini Falajs, facilitating the identification of areas with noteworthy population

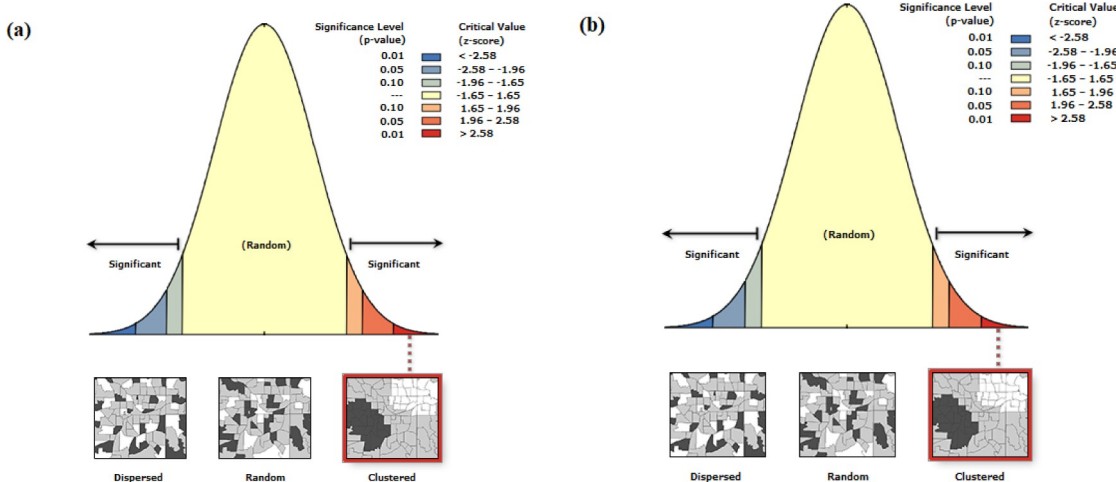

**Fig 3. Shows the spatial distribution patterns of live (active) Aini with a z-score of 16.43, indicating that there is a less than 1% possibility that this clustered pattern is the product of random chance (Fig 3A).** Spatial distribution patterns of dried (non-functional) Aini Falajs with a z-score of 53.09, indicating that there is a less than 1% possibility that this clustered pattern is the product of random chance (Esri ArcGIS Pro 2023).

abundance for further investigation. Fig 4 visually depicts hotspots and cold spots, significantly contributing to our understanding of Aini Falaj distribution and spatial dynamics within the study area (Fig 4). This information sheds light on their ecological preferences and potential habitat associations, aiding in conservation strategies and targeted studies.

The KD analysis revealed significantly high-density areas of Aini Falajs in the northern regions of Al-Dakhiliyah and Al-Sharqiyah North Governorates, as well as the southern

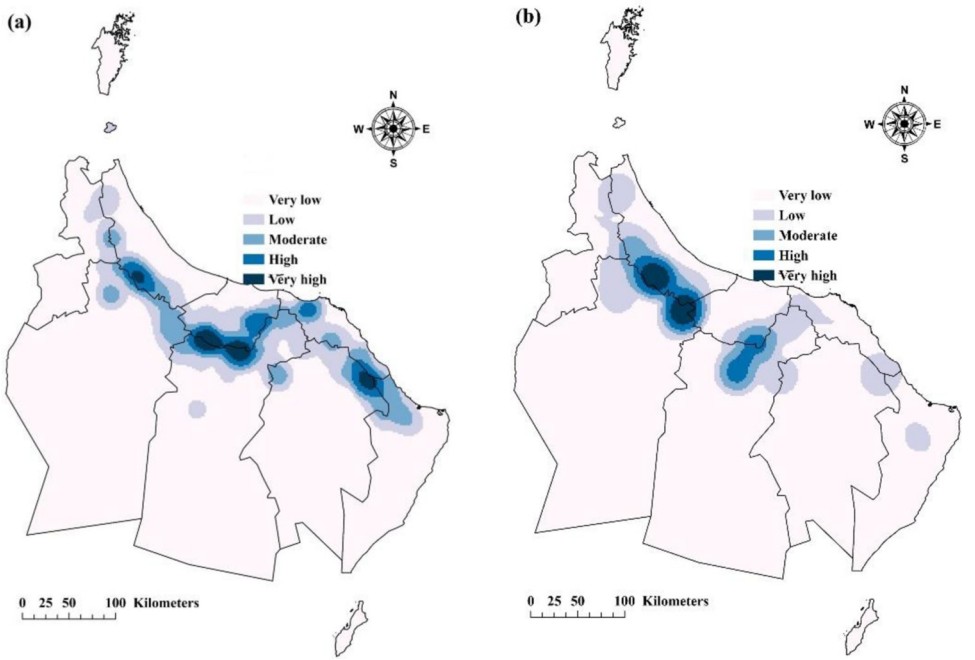

**Fig 4.** Depicts the KD analysis, which showcases the distribution of live (Fig 4A) and dried (Fig 4B) Aini Falajs in the study area by highlighting hotspots and cold spots (Esri ArcGIS Pro 2023).

regions of Al-Batinah North and Al-Batinah North Governorates for live Aini Falajs (Fig 4A). High-density areas of non-functional Aini Falajs were observed in the southern regions of Al-Batinah North and the northern areas of Al-Dhahirah Governorates (Fig 4B). These findings indicate localized clusters of Aini Falajs, suggesting favorable environmental conditions or historical factors contributing to their concentrated distribution in specific areas. The identification of these high-density zones provides valuable information for understanding spatial patterns and potential factors influencing the prevalence and sustainability of Aini Falajs in the study area.

## MGWR

The MGWR models employed in this study yielded significant results. In the live status model, approximately 99.5% of the variability in the live status of Aini Falajs could be explained by the considered environmental factors (R-squared = 0.995). This strong relationship was further supported by the adjusted R-squared value of 0.993, indicating the robustness of the associations between the predictors and the live status. The AICc value of -1532.29 suggested an optimal balance between model complexity and fit. Additionally, the sigma-squared estimate of 0.0066 and MLE estimate of 0.0041 demonstrated the model's proficiency in capturing observed variability. The incorporation of effective degrees of freedom (663.05) accommodated spatial heterogeneity in the relationships.

Similarly, in the dead status model, the MGWR analysis showed excellent goodness-of-fit. The environmental factors accounted for approximately 99.8% of the variability in the dead status (R-squared = 0.998). The adjusted R-squared value of 0.997 further validated the model's effectiveness. The model's fit was strongly supported by the AICc value of -2767.0483. The model's capacity to capture observed variability was validated by the sigma-squared estimate of 0.0022 and the MLE estimate of 0.0015. The effective degrees of freedom (688.92) accounted for the associations' geographical variability. These findings illustrate the substantial links between the environmental elements studied and Aini Falajs's standing. The strong R-squared values suggest that the models can explain a significant percentage of the observed variability. The AICc values indicate that the models strike a good balance of complexity and fit. The sigma-squared values show how well the models capture observed variability, and the inclusion of effective degrees of freedom allows for regional heterogeneity. Overall, our data shed light on the regional distributions and variables determining the live and dead status of Aini Falajs in the studied area.

Fig 4 depicts the distribution and intensity of the correlations between the live and dry Aini Falajs and environmental parameters. These maps provide useful information on the impact of environmental variables on falajs throughout the research area. The maps portray the intensity and direction of the correlations through the use of colors and patterns, allowing for a clearer comprehension of the spatial patterns and linkages between the falajs and environmental factors.

To analyze the strength of the linkages driving the geographical distribution of the living Aini Falajs, $R^2$ values were obtained (Fig 5A). The $R^2$ values found varied from 0.07 to 0.99, demonstrating a wide range of relationship. The mean $R^2$ value of 0.36 and the median $R^2$ value of 0.32 show the average and central trends of these connections, respectively. Notably, the standard deviation of 0.19 indicates that the strength of the correlations detected across the research region varies. Similarly, the computed $R^2$ values for dry Aini Falaj ranged from 0.08 to 0.99. The mean $R^2$ value of 0.36 and the median of 0.32 mirror the findings for the live Aini Falajs, suggesting comparable levels of association with the environmental factors (Fig 5B).

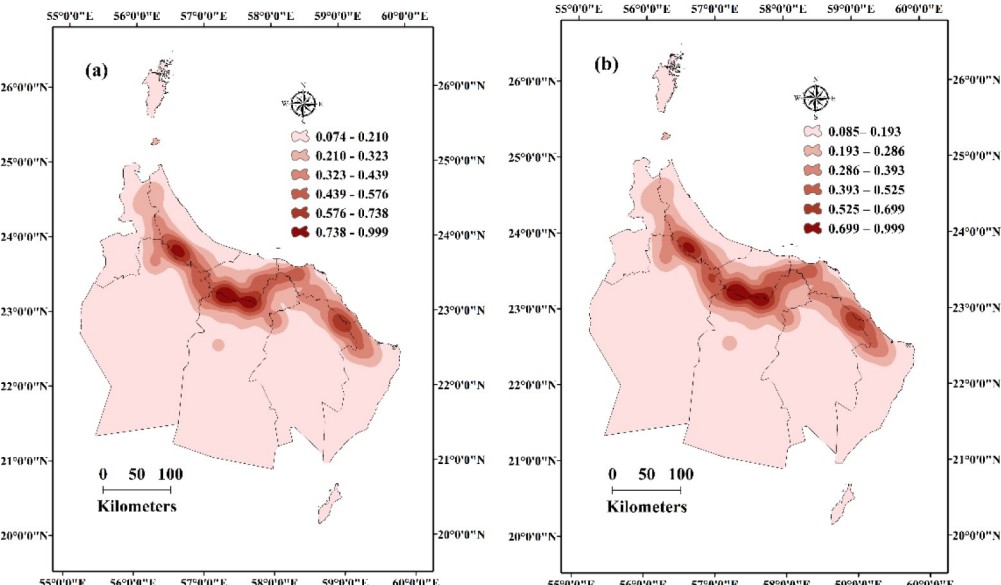

**Fig 5.** Illustrates the relationships between the densities of live Falaj (Fig 5a), dried Falajs (Fig 5b), and environmental factors, as determined by the $R^2$ values (Esri ArcGIS Pro 2023).

The standard deviation of 0.19 further emphasizes the presence of variability in the strength of these relationships.

These findings underscore the significance of the selected environmental factors in shaping the distribution and status of both the live and dry Aini Falajs within the study area. The $R^2$ values, complemented by the results maps, offer comprehensive insights into the intricate interplay between the falajs and the environmental variables.

**Table 2. Overall results of the spatial relationships between alive of Aini Falajs and environmental factors in the study area.**

| Variables | Mean | Standard Deviation | Minium | Median | Maximum |
|---|---|---|---|---|---|
| interception | 0.189 | 0.6294 | -0.775 | 0.2126 | 1.375 |
| Evaporation | -0.1947 | 0.3253 | -0.7 | -0.3838 | 1.0619 |
| Solar radiation | -0.1387 | 0.5395 | -0.2727604 | -0.101 | 1.667 |
| Precipitation | 0.02 | 0.1801 | -0.3924 | 0.0373 | 0.7807 |
| VC | 0.0025 | 0.0002 | 0.0021 | 0.0026 | 0.003 |
| Distance to drainages | -0.0022 | 0.0191 | -0.0789 | -0.005 | 0.1307 |
| Drainages density | -0.1168 | 0.2211 | -1.3516 | 0.1311 | 1.4484 |
| TWI | 0.0098 | 0.0066 | -0.0007 | 0.008 | 0.0222 |
| SPI | -0.0014 | 0.0005 | -0.002 | -0.0016 | -0.0005 |
| Slope Aspect | 0.0023 | 0.0004 | 0.0017 | 0.0022 | 0.0029 |
| Aspect | 0.005 | 0.0425 | -0.178 | 0.0016 | 0.3418 |
| Slope | 0.0086 | 0.0002 | 0.0083 | 0.0005 | 0.0089 |
| Soil types | -0.0038 | 0.0145 | -0.035 | -0.0031 | 0.0166 |
| Plan curvature | 0.0103 | 0.0021 | 0.0069 | 0.0106 | 0.013 |
| Profile curvature | 0.0034 | 0.0004 | 0.003 | 0.0032 | 0.0042 |
| Geological | 0.064 | 0.1082 | -0.3958 | 0.0507 | 0.6976 |
| TSI | -0.0017 | 0.0519 | -0.1864 | -0.0034 | 0.1527 |
| Agricultural soil | -0.1859 | 0.4924 | -1.8102 | -0.0244 | 0.6998 |

## Relationships between the alive of Aini Falajs and environmental factors

The results of the MGWR analysis revealed significant associations between the spatial distribution of active Aini Falajs and various environmental factors. Of the examined variables, precipitation exhibited a slight positive impact, suggesting the importance of water availability for the thriving of Aini Falaj systems. The mean precipitation value was 0.02 (SD = 0.18) with a range of -0.39 to 0.78, indicating that areas with higher interception and precipitation levels may support more vibrant Aini Falaj systems (Table 2 and Fig 6A). Conversely, variables such as evaporation and solar radiation exhibited negative influences on the live status of the falajs. The mean value for evaporation was -0.19 (SD = 0.35) ranging from -0.70 to 1.16, implying that higher evaporation rates have a detrimental impact on the sustainability of the falajs (Table 2 and Fig 6B). Similarly, solar radiation demonstrated a mean value of -0.14 (SD = 0.54) with a range of -0.28 to 1.70, indicating that excessive solar radiation negatively affects the falaj systems (Table 2 and Fig 6C).

Furthermore, variables related to distance to drains and drains density also displayed negative influences. The mean value for distance to drains was -0.02 (SD = 0.019), ranging from -0.079 to 0.13, suggesting that closer proximity to drains hinders the health of the falajs (Table 2 and Fig 6D). Drains density exhibited a mean value of -0.12 (SD = 0.22) with a range of -1.35 to 1.49, indicating that a higher density of drains has adverse effects on the sustainability and vitality of the falaj systems (Table 2 and Fig 6E).

Moreover, variables such as topographic wetness index (TWI), aspect, and slope demonstrated positive influences on the live status of the falajs. TWI exhibited a mean value of 0.09 (SD = 0.01) with a range of -0.007 to 0.022, indicating that areas with higher values of TWI are associated with healthier falajs (Table 2 and Fig 6F). Similarly, aspect and slope exhibited positive means, with aspect having a mean of 0.005 (SD = 0.04) and a range of -0.18 to 0.34 (Table 2 and Fig 6G), and slope having a mean of 0.086 (SD = 0.002) with a range of 0.0038 to 0.0089 (Table 2 and Fig 6H). These results suggest that steeper slopes and specific aspects contribute to the vitality and sustainability of the falaj systems. In addition, the average slope aspect value was 0.0023 (standard deviation = 0.0004) (Table 2 and Fig 6I). Understanding slope aspect factor is critical because it effects solar exposure and water runoff patterns. Aini Falajs may benefit from areas with certain slope features. Slope aspect may be used to aid management and preservation of falaj systems by maximizing light and moisture conditions.

The mean value of agricultural soil was -0.1859, showing a general negative influence on the falajs' life status. The standard deviation of 0.4924 indicates significant variation in the distribution of agricultural soil across the research region. The minimum recorded value of -1.8102 and maximum recorded value of 0.6998 further demonstrate the variety of agricultural soil values observed (Table 2 and Fig 6j). These findings imply that locations with a larger share of agricultural soil may have a negative impact on the sustainability and vitality of the Aini Falaj systems. Agricultural activities and the composition of soil types can have an impact on the ecological dynamics and health of the falajs. The mean value for soil types was -0.0038 (SD = 0.0145). This suggests that specific soil types may have a slight negative impact on the live status of the Aini Falajs (Table 2 and Fig 6K). Understanding the distribution and characteristics of different soil types can inform land management practices to preserve and enhance the vitality of the falajs.

Geological factors had a mean value of 0.064 and a standard deviation of 0.1082, according to the research (Table 2 and Fig 6L). These findings imply that specific geological factors contribute to the Aini Falajs' vitality. Soil qualities, water movement, and general ecosystem dynamics can all be influenced by geological elements such as rock types, structural features, and mineral content. Areas with good geological characteristics may offer better support for the falajs' sustainability and health.

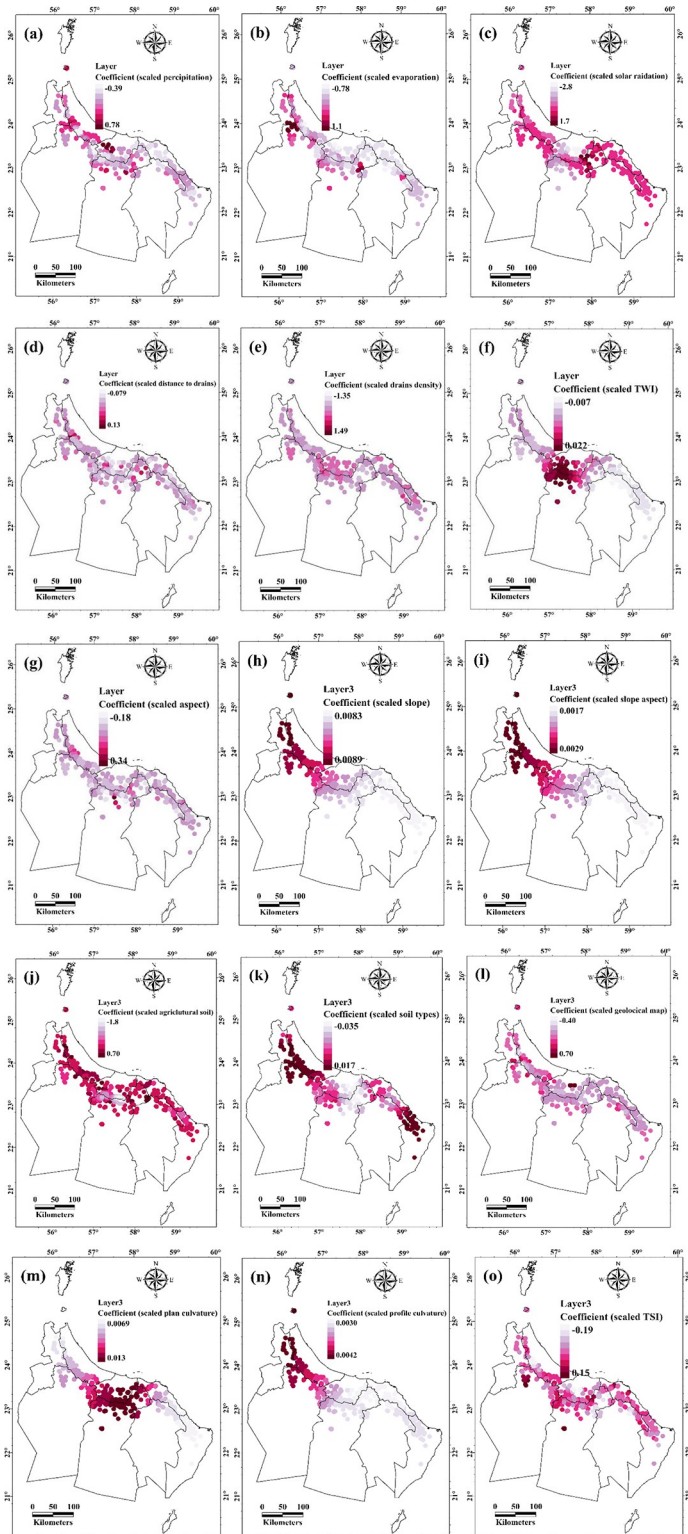

**Fig 6. Coefficients illustrate the spatial relationships between alive of Aini Falajs and environmental factors (Esri ArcGIS Pro 2023).**

The mean plan curvature value was 0.0103 (SD = 0.0021). This suggests that areas with higher plan curvature values are associated with more vibrant Aini Falajs. Plan curvature, representing the curvature of the land surface in the horizontal plane, influences water accumulation and drainage patterns, contributing to the falajs' health and functionality (Table 2 and Fig 6M). The mean profile curvature value was 0.0034 (SD = 0.0004). Areas with higher profile curvature values may positively impact the live status of the Aini Falajs (Table 2 and Fig 6N). Profile curvature, representing the curvature of the land surface in the vertical plane, affects water movement, soil moisture distribution, and vegetation patterns, contributing to the falajs' vitality and sustainability.

The TSI had a mean of -0.0017 and a standard deviation of 0.0519. Lower TSI levels may have a modest negative impact on the viability of the Aini Falajs. TSI is a measure of a location's relative position within its surrounding terrain, which influences water availability and soil moisture. The vegetation cover in the Aini Falajs has a mean value of 0.0025 (SD = 0.0002), ranging from 0.0021 to 0.003 (Fig 6O). These findings imply a continuous and persistent presence of vegetation across the research region. The small range of results reflects a homogeneous vegetative cover over the falajs. Maintaining this vegetative cover is critical for preserving biodiversity, reducing soil erosion, and sustaining the general health and sustainability of the Aini Falajs (S1 File).

## The relationships between the dead of Aini Falajs and environmental factors

In contrast, the MGWR analysis revealed associations between the spatial distribution of the dry Aini Falajs and various environmental factors. Variables such as evaporation and solar radiation displayed negative influences on the active Falajs. Evaporation exhibited a mean value of -0.0204 (SD = 0.22) and ranged from -0.85 to 0.81, suggesting that higher evaporation rates may contribute to the demise of the falaj systems (Table 3 and Fig 7A). Similarly, solar

**Table 3. Overall results of the spatial relationships between dead of Aini Falajs and environmental factors in the study area.**

| Variables | mean | Standard Deviation | Minium | Median | Maximum |
|---|---|---|---|---|---|
| interception | 0.189 | 0.6294 | -0.775 | 0.2126 | 1.375 |
| Evaporation | -0.1947 | 0.3253 | -0.7 | -0.3838 | 1.0619 |
| Solar radiation | -0.1387 | 0.5395 | -0.2727604 | -0.101 | 1.667 |
| Precipitation | 0.02 | 0.1801 | -0.3924 | 0.0373 | 0.7807 |
| VC | 0.0025 | 0.0002 | 0.0021 | 0.0026 | 0.003 |
| Distance to drainages | -0.0022 | 0.0191 | -0.0789 | -0.005 | 0.1307 |
| Drainage density | -0.1168 | 0.2211 | -1.3516 | 0.1311 | 1.4484 |
| TWI | 0.0098 | 0.0066 | -0.0007 | 0.008 | 0.0222 |
| SPI | -0.0014 | 0.0005 | -0.002 | -0.0016 | -0.0005 |
| Slope Aspect | 0.0023 | 0.0004 | 0.0017 | 0.0022 | 0.0029 |
| Aspect | 0.005 | 0.0425 | -0.178 | 0.0016 | 0.3418 |
| Slope | 0.0086 | 0.0002 | 0.0083 | 0.0005 | 0.0089 |
| Soil types | -0.0038 | 0.0145 | -0.035 | -0.0031 | 0.0166 |
| Plan curvature | 0.0103 | 0.0021 | 0.0069 | 0.0106 | 0.013 |
| Profile curvature | 0.0034 | 0.0004 | 0.003 | 0.0032 | 0.0042 |
| Geological | 0.064 | 0.1082 | -0.3958 | 0.0507 | 0.6976 |
| TSI | -0.0017 | 0.0519 | -0.1864 | -0.0034 | 0.1527 |
| Agricultural soil | -0.1859 | 0.4924 | -1.8102 | -0.0244 | 0.6998 |

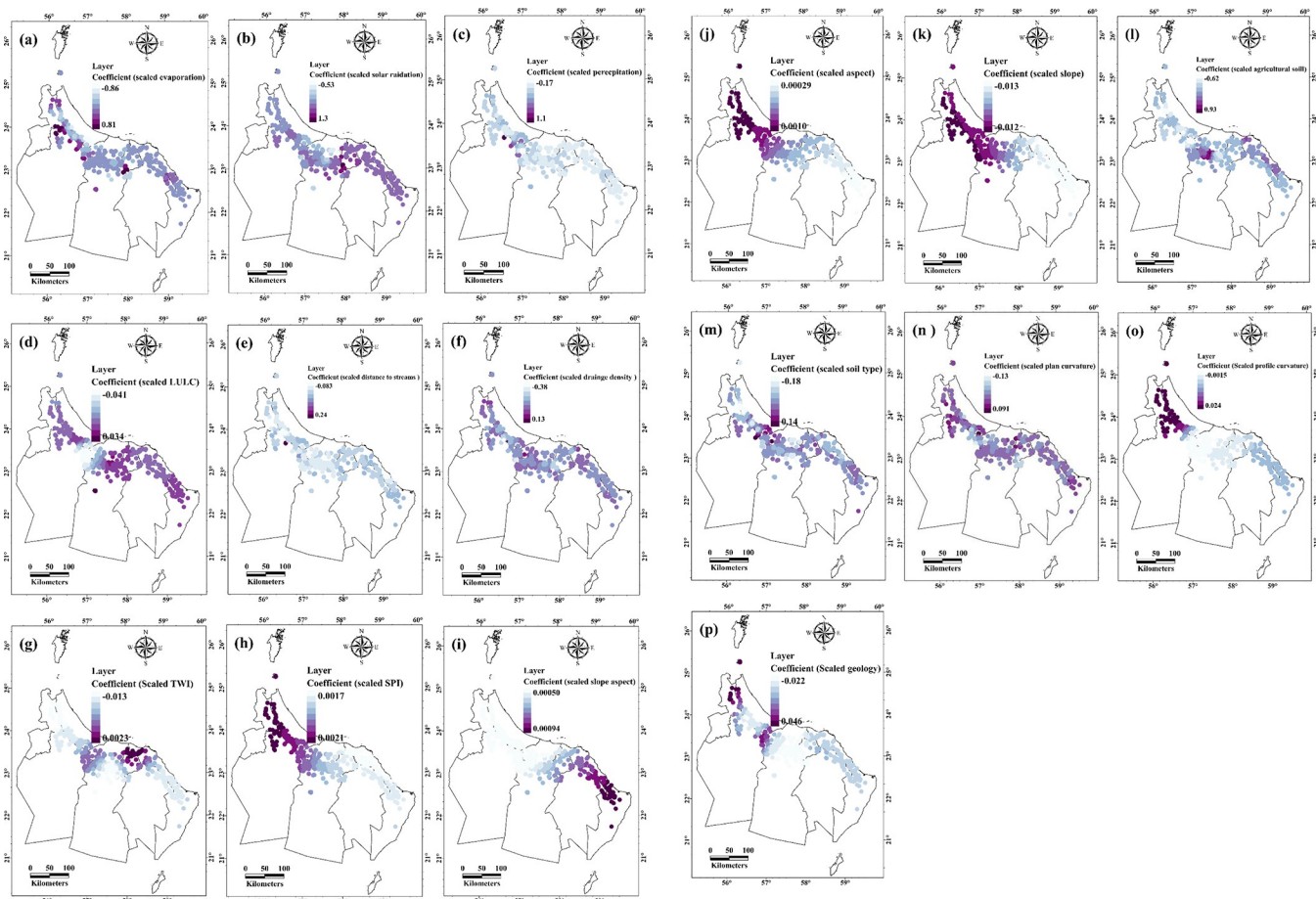

**Fig 7. Coefficients illustrate the spatial relationships between death of Aini Falajs and environmental factors (Esri ArcGIS Pro 2023).**

radiation demonstrated a mean value of 0.48 (SD = 0.186) and ranged from -0.53 to 1.26, indicating that excessive solar radiation can have detrimental effects (Table 3 and Fig 7B). These findings imply that controlling evaporation rates and providing adequate shading mechanisms may be crucial in maintaining the live status of the falaj systems. Precipitation, with a mean of 0.1373, indicating a positive association with the death of AFs. Its standard deviation of 0.1072 indicates relatively minimal variability. The minimum value of -0.17 and highest value of 1.1382 show a range of probable connections with the variable of interest (Table 3 and Fig 7C). The mean value of vegetation cover is 0.0047, indicating a tiny positive average connection with AF death. The standard deviation of 0.0138 suggests that there is moderate variability around the mean, meaning that the strength of the link may vary between data or locations. The minimum value of 0.04 and maximum value of 0.0337 demonstrate the dataset's limited range of correlations (Table 3 and Fig 7D).

Furthermore, variables such as distance to drains and drains density also displayed negative influences on the falajs' status. The mean value for distance to drains was 0.0021 (SD = 0.0267) and ranged from -0.08 to 0.24, implying that closer proximity to drains may contribute to the decline of the falaj systems (Table 3 and Fig 7E). Similarly, drains density exhibited a mean value of -0.011 (SD = 0.047) and ranged from -0.38 to 0.252, suggesting that higher density of drains may have adverse effects on their sustainability and vitality (Table 3 and Fig 7F). These findings highlight the importance of proper drainage planning and management to prevent

the transition of falajs from live to dead status. The TWI shows a weak or negligible relationship with the variable "dead of AFs." The mean value of -0.007 suggests a slightly negative association, indicating that higher TWI values (indicating wetter areas) may be associated with a slightly lower occurrence of the variable (Table 3 and Fig 7G). The average SPI value was 0.0019, indicating a near-zero average connection with the death of AFs. This means that there is minimal departure from typical precipitation circumstances in regard to the variable of interest on average. The standard deviation of 0.0001 suggests that there is little variation around the mean, showing that the observed relationships are quite consistent throughout the sample (Table 3 and Fig 7H). Slope aspect, which indicates the direction of slopes, shows a nearly neutral average association with the distribution of death of Aini Falajs with a mean of 0.0006. The standard deviation of 0.0001 suggests low variability around the mean. The minimum value of 0.0017 indicates the presence of relatively steep slope aspects, while the maximum value of 0.0009 suggests the presence of more gentle slopes (Table 3 and Fig 7I).

Moreover, variables related to soil types and agricultural practices demonstrated associations with the dead status of Aini Falajs. Soil types exhibited a mean value of -0.03 (SD = 0.015) and ranged from -0.18 to 0.15, indicating that specific soil characteristics may contribute to the deterioration of the falaj systems (Table 3 and Fig 7J). Additionally, agricultural soil demonstrated a mean value of 0.0415 (SD = 0.2755) and ranged from -0.62 to 0.93, suggesting that certain agricultural practices may have negative impacts on the falajs' health (Table 3 and Fig 7K). These findings emphasize the need for sustainable land-use practices and soil conservation strategies to preserve the live status of the falaj systems (S2 File).

## Evolution the MGWR models

In the context of our study on Aini Falajs, the Spatial Autocorrelation analysis was conducted on both live and dead Aini Falajs' regression residuals. The absence of statistically significant clustering in the residuals indicates that the regression models were accurately defined. The study yielded a non-significant result for the live Aini Falajs, showing negligible spatial autocorrelation (Moran's Index: -0.0008, Z-score: 0.0273, P-value: 0.9781). Similarly, the dead Aini Falajs' investigation revealed a modest and non-significant negative spatial autocorrelation (Moran's Index: -0.0029, Z-score: -0.5937, P-value: 0.5526). These findings suggest that the MGWR models employed in our study are appropriate and effective in capturing the spatial relationships within the data. The absence of significant clustering in the residuals supports the notion that the models adequately account for spatial dependence and patterns. Therefore, the regression models can be deemed suitable for further analysis and interpretation.

## Discussion

The current study sought to evaluate the environmental parameters that influence the distribution and appropriateness of Aini Falajs in the study area. Collinearity diagnostics, spatial autocorrelation analysis, KD mapping, and MGWR approaches were all used in the analysis. Collinearity diagnostics were used to investigate the interrelationships between the independent environmental variables impacting Aini Falajs. The findings revealed no substantial multicollinearity problems, as the majority of the variables had Variance Inflation Factors (VIFs) below the critical threshold of 5 [50]. Variables such as evaporation, solar radiation, and precipitation exhibited moderate levels of association, whereas others showed little intercorrelation. The absence of multicollinearity problems suggests that each variable contributes individually to the analysis and may be securely used in further modeling or regression investigations.

Spatial autocorrelation analysis revealed a different clustering pattern for both active and dry Aini Falajs distributions in the study area. The presence of spatial dependence was revealed by the positive Moran's Index value, which indicated that neighboring Falajs exhibited comparable characteristics. This clustering pattern shows that underlying geographical processes or environmental factors influence how Aini Falajs are arranged and distributed. The findings highlight the importance of considering spatial links, as well as the potential influence of environmental factors on the distribution of these groundwater features [51, 52].

The hotspots and coldspots, which represent locations with high and low concentrations of both alive and deceased falajs, respectively, are well highlighted by the KD maps (see Fig 3A and 3B). Hotspots are heavily populated groupings of falajs with higher densities than the surrounding areas. These hotspots provide useful information on areas of increased aggregation and large Aini Falajs population abundance. Coldspots, on the other hand, depict areas where falajs are less densely dispersed, indicating their rarity or infrequency.

The KD analysis revealed significantly high densities of alive Aini Falajs in the northern regions of Al-Dakhiliyah and Al-Sharqiyah North Governorates, as well as in the southern regions of Al-Batinah North and Al-Batinah North Governorates (see Fig 3A). For deceased falajs, high-density areas were observed in the southern regions of Al-Batinah North and the northern areas of Al-Dhahirah Governorates (see Fig 3B). These findings indicate localized clusters of falajs, suggesting the presence of favorable environmental conditions or historical factors contributing to their concentrated distribution in these particular areas. The identification of these high-density zones provides valuable information for understanding the spatial patterns and potential factors influencing the prevalence and sustainability of Aini Falajs in the study area. Sahour et al. [53], observed similarly to our work, that spatial pattern analysis is a valuable tool for analyzing the distribution of groundwater resources and the variables influencing groundwater quality and quantity.

The findings of the MGWR study give deep insights into the complex relationships between Aini Falajs' live and dead statuses and a number of environmental variables. The study revealed numerous significant discoveries for the live Aini Falajs. The findings of the MGWR study give deep insights into the complex relationships between Aini Falajs' live and dead statuses and a number of environmental variables. The study revealed numerous significant discoveries for the live Aini Falajs. Variables such as precipitation, and slope exhibit positive influences, indicating their beneficial impact on the live status of the falajs. Higher levels of precipitation enhance water availability, thereby fostering a conducive environment for the sustenance of the falaj systems [54]. Furthermore, steeper slopes contribute to greater drainage and water movement, which supports the falajs' healthy nature. Variables such as evaporation, sun radiation, distance to drains, and drain density, on the other hand, have a detrimental impact on the falajs' inhabiting state. Elevated evaporation rates and intense sun radiation may result in water loss and increased moisture stress, impairing the falajs' viability. Similar to our study, several studies found that evaporation variable influence discharge of the aflaj systems [55, 56].

According to our findings, being close to drains and having a high density of drains may have negative effects on natural hydrological dynamics and falaj sustenance. As a result, clever management techniques such as water conservation, shading devices, and correct drainage systems are crucial for preserving the resilience and vitality of the living Aini Falajs.

In contrast, our study offers insight on the environmental elements connected with Aini Falajs' death status. Variables such as sun radiation, precipitation, and profile curvature have positive effects on the dead status, indicating that they contribute to the falajs' decline and deterioration. Intensified solar radiation can cause increased evapotranspiration, increasing water scarcity difficulties and threatening the survival of falajs. Higher precipitation levels,

although seemingly beneficial, may indicate extreme events or mismanagement, resulting in waterlogging, reduced oxygen availability, and the subsequent demise of the falajs [57]. Moreover, profile curvature, representing the landform characteristics, displays a positive influence on the dead status, suggesting that irregular topography and suboptimal land conditions may contribute to the falajs' decline.

Variables such as evaporation, distance to drains, and agricultural soil exhibit minimal influences on the dead status, implying that their impacts may be relatively insignificant in the context of falaj degradation. However, it is essential to recognize their potential contributions to the overall health and sustainability of the falajs, albeit with lesser prominence compared to other influential factors. Therefore, strategic interventions targeting solar radiation management, sustainable precipitation regimes, landform improvements, and comprehensive land-use practices hold promise in reviving the dead Aini Falajs and restoring their optimal functioning state.

The in-depth examination of these environmental elements provides crucial insights into the complex dynamics governing both the currently alive and deceased Aini Falajs. The findings emphasize the significance of holistic and sustainable management measures that include water conservation, shading mechanisms, proper drainage systems, and land-use planning to preserve the falaj systems' vibrancy, resilience, and long-term survival. As a result, comparative research on the spatial distribution and environmental determinants of the Aini Falajs on a wide scale are limited. Several research, however, have addressed related issues or provided important insights [58–60]. For example, Buerkert et al. [61], investigated the hydrogeological characteristics of Aini Falajs in Al Jabal Al Akhdar, northern Oman, providing a foundation for understanding their occurrence and behavior. While these studies may not cover every aspect of Aini Falajs, they do add to our understanding of these groundwater characteristics.

The current study contributes to our understanding of the spatial distribution, environmental appropriateness, and potential affecting factors of Aini Falajs in the studied area. The absence of substantial multicollinearity problems among environmental variables assures that each variable contributes uniquely to subsequent analyses. The observed clustering pattern and hotspots detected by spatial autocorrelation and KD mapping emphasize the importance of spatial linkages and probable environmental variables driving the distribution of Aini Falajs. The identification of high-density areas in specific regions shows the presence of favorable environmental circumstances or historical factors that contribute to their concentrated distribution.

It is worth noting that comprehensive studies specifically focused on Aini Falajs at a large scale are limited. The scarcity of such studies underscores the novelty and significance of this research, as it provides valuable insights into the distribution and influencing factors of these unique groundwater features. The Aini Falaj system represents an important cultural and hydrological heritage in the Sultanate of Oman, and further research is necessary to fully comprehend their characteristics, behavior, and environmental significance.

## Conclusion

In conclusion, this comprehensive study provides significant insights into the distribution, status, and influencing factors of Aini Falajs in the study area. The examination of several environmental factors exposes their separate contributions and highlights the lack of considerable multicollinearity. The MGWR models' findings show outstanding goodness-of-fit, with high R-squared values suggesting the models' significant explanatory power for both live and dead falajs. The usage of KD maps gives excellent visual portrayals of Aini Falajs distribution patterns, highlighting hotspots and coldspots. These geographical patterns highlight areas of high

and low densities, suggesting areas of concentrated population abundance and relative paucity of falajs, respectively. The discovery of these hotspots and coldspots assists in understanding the ecological preferences and possible habitat linkages of Aini Falajs, allowing for focused conservation initiatives and additional research. The work fills a huge research vacuum by offering thorough insights on Aini Falajs, as there have been few large-scale studies especially focused on these distinctive groundwater characteristics. The findings add to a better knowledge of groundwater dynamics in Oman, where geological properties, topography, and hydrological connection have been highlighted as critical determinants. This study improves our understanding of the distribution and sustainability of Aini Falajs by taking these aspects into account. This work is significant not just for its scholarly contributions, but also for its practical consequences. The findings can help guide evidence-based decision-making in the Sultanate of Oman for the preservation, management, and conservation of Aini Falajs. The study emphasizes the necessity of ongoing research and monitoring efforts to fully understand the features, behavior, and environmental significance of these falajs. As a result, this study contributes to our understanding of Aini Falajs by offering useful insights into their distribution, status, and affecting variables. The strong statistical models, together with the spatial analysis utilizing KD maps, help to our knowledge of the biological dynamics and conservation requirements of these unique groundwater features. These findings have implications for sustainable water resource management and cultural preservation in the study region, as well as laying the groundwork for future research and conservation initiatives.

## Supporting information

**S1 File. Demonstrates the outcomes of impact and Cook's D values.** Both influence and Cook's D values assess the impact of a feature on the estimate of regression coefficients, considering the vitality of Aini Falajs and environmental factors.
(PDF)

**S2 File. Illustrates the results of influence and Cook's D values.** Both of and influence Cook's D values measure the influence of the feature on the estimation of the regression coefficients the dead of Aini Falajs and environmental factors.
(PDF)

## Acknowledgments

We are grateful to The Ministry of Agriculture, Fisheries Wealth and Water Resources, Oman, for their support and for providing the data used in this study.

## Author Contributions

**Conceptualization:** Khalifa M. Al-Kindi.

**Data curation:** Khalifa M. Al-Kindi.

**Formal analysis:** Khalifa M. Al-Kindi.

**Funding acquisition:** Khalifa M. Al-Kindi.

**Investigation:** Khalifa M. Al-Kindi.

**Methodology:** Khalifa M. Al-Kindi.

**Project administration:** Khalifa M. Al-Kindi.

**Resources:** Khalifa M. Al-Kindi.

**Software:** Khalifa M. Al-Kindi.

**Supervision:** Khalifa M. Al-Kindi.

**Validation:** Khalifa M. Al-Kindi.

**Visualization:** Khalifa M. Al-Kindi.

**Writing – original draft:** Khalifa M. Al-Kindi.

**Writing – review & editing:** Khalifa M. Al-Kindi.

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
