## [Decision Letter · Decision Letter 0]

10 Dec 2023

PONE-D-23-30564Assessing the Environmental Factors Affecting the Sustainability of Ayni Falaj SystemPLOS ONE

Dear Dr. Alkindi,

Thank you for submitting your manuscript to PLOS ONE. After careful consideration, we feel that it has merit but does not fully meet PLOS ONE’s publication criteria as it currently stands. Therefore, we invite you to submit a revised version of the manuscript that addresses the points raised during the review process.

We look forward to receiving your revised manuscript.

Kind regards,

Abdulkader Murad, Ph.D

Academic Editor

PLOS ONE

5. We note that you have stated that you will provide repository information for your data at acceptance. Should your manuscript be accepted for publication, we will hold it until you provide the relevant accession numbers or DOIs necessary to access your data. If you wish to make changes to your Data Availability statement, please describe these changes in your cover letter and we will update your Data Availability statement to reflect the information you provide

6. Please include a separate caption for each figure in your manuscript

7. We note that Figures 1,2,4,5,6 and 7 in your submission contain [map/satellite] images which may be copyrighted. All PLOS content is published under the Creative Commons Attribution License (CC BY 4.0), which means that the manuscript, images, and Supporting Information files will be freely available online, and any third party is permitted to access, download, copy, distribute, and use these materials in any way, even commercially, with proper attribution. For these reasons, we cannot publish previously copyrighted maps or satellite images created using proprietary data, such as Google software (Google Maps, Street View, and Earth). For more information, see our copyright guidelines: http://journals.plos.org/plosone/s/licenses-and-copyright.

a. You may seek permission from the original copyright holder of Figures 1,2,4,5,6 and 7 to publish the content specifically under the CC BY 4.0 license. 

Additional Editor Comments:

Reviewer 1

RE: Assessing the Environmental Factors Affecting the Sustainability of Ayni Falaj System

This manuscript is well prepared and of high quality. In this article, the order of presenting the content is well respected and innovative. Therefore, I consider this manuscript suitable for acceptance for publication in the journal. However, before acceptance, minor revisions are required as follows:

1- It is better to separate the section related to the study area in the materials and methods section and put it before the materials and methods section.

2- It is better to edit the text accurately by an English speaker.

Reviewer 2

The article Assessing the Environmental Factors Affecting the Sustainability of Ayni Falaj

System has been read in its entirety by this reviewer and highlighted areas indicate the

locations of my following comments. It is important to note that my specialty is in multivariate

GIS as it pertains to ecology, however I am very familiar with the software and process

described in this paper as I use them on a regular basis. I have found its use of GIS processes

appropriate, innovative, and enlightening. Please see the following comments as suggestions to

improve the article's readability and specificity.

Comments:

Lines 38 - 40: These two sentences appear redundant and could be shortened to a single

sentence.

Line 49: The situation described does not have to be specific to a country and "within the

country" could be dropped.

Line 53(1): "lives" could be replaced with "way of life"

Line 53(2): "urbanization and" falls under the category of changes to land use and could be

dropped from the sentence.

Line 56: "this" may be dropped from the sentence.

Line 58: "confronts" feels inappropriate may be replaced with "faces".

Line 189: This sentence implies the AFS is impacting the environmental variables, not the other

way around which is what I think you meant.

Line 382: "Evaporation" looks to be accidently bolded.

Line 407: "compared" may be replaced with "in relation"

Line 451: The word "successfully" appears out of place as it is making a suggestion to the

reader's interpretation and could be dropped from the sentence.

In summary I found this article to be a novel and powerful use of multivariate GIS

analysis. I believe the findings are strong and honest. Figures including the maps are

appropriate and readable. Thank you for considering my comments.

Reviewers' comments:

Reviewer's Responses to Questions

**Comments to the Author**

1. Is the manuscript technically sound, and do the data support the conclusions?

Reviewer #1: Yes

Reviewer #2: Yes

2. Has the statistical analysis been performed appropriately and rigorously? 

Reviewer #1: Yes

Reviewer #2: Yes

3. Have the authors made all data underlying the findings in their manuscript fully available?

Reviewer #1: Yes

Reviewer #2: Yes

4. Is the manuscript presented in an intelligible fashion and written in standard English?

Reviewer #1: No

Reviewer #2: Yes

5. Review Comments to the Author

Reviewer #1: RE: Assessing the Environmental Factors Affecting the Sustainability of Ayni Falaj System

This manuscript is well prepared and of high quality. In this article, the order of presenting the content is well respected and innovative. Therefore, I consider this manuscript suitable for acceptance for publication in the journal. However, before acceptance, minor revisions are required as follows:

1- It is better to separate the section related to the study area in the materials and methods section and put it before the materials and methods section.

2- It is better to edit the text accurately by an English speaker.

Reviewer #2: The article Assessing the Environmental Factors Affecting the Sustainability of Ayni Falaj

System has been read in its entirety by this reviewer and highlighted areas indicate the

locations of my following comments. It is important to note that my specialty is in multivariate

GIS as it pertains to ecology, however I am very familiar with the software and process

described in this paper as I use them on a regular basis. I have found its use of GIS processes

appropriate, innovative, and enlightening. Please see the following comments as suggestions to

improve the article's readability and specificity.

Comments:

Lines 38 - 40: These two sentences appear redundant and could be shortened to a single

sentence.

Line 49: The situation described does not have to be specific to a country and "within the

country" could be dropped.

Line 53(1): "lives" could be replaced with "way of life"

Line 53(2): "urbanization and" falls under the category of changes to land use and could be

dropped from the sentence.

Line 56: "this" may be dropped from the sentence.

Line 58: "confronts" feels inappropriate may be replaced with "faces".

Line 189: This sentence implies the AFS is impacting the environmental variables, not the other

way around which is what I think you meant.

Line 382: "Evaporation" looks to be accidently bolded.

Line 407: "compared" may be replaced with "in relation"

Line 451: The word "successfully" appears out of place as it is making a suggestion to the

reader's interpretation and could be dropped from the sentence.

In summary I found this article to be a novel and powerful use of multivariate GIS

analysis. I believe the findings are strong and honest. Figures including the maps are

appropriate and readable. Thank you for considering my comments.

6. PLOS authors have the option to publish the peer review history of their article (what does this mean?). If published, this will include your full peer review and any attached files.

Reviewer #1: No

Reviewer #2: **Yes: **Kevin Maxwell Lester

---

## [Author Response · Author response to Decision Letter 0]

26 Dec 2023

Reviewer # 1

RE: Assessing the Environmental Factors Affecting the Sustainability of Ayni Falaj System

This manuscript is well prepared and of high quality. In this article, the order of presenting the content is well respected and innovative. Therefore, I consider this manuscript suitable for acceptance for publication in the journal. However, before acceptance, minor revisions are required as follows:

Dear Reviewer, 

Thank you so much for your thorough and constructive feedback on our manuscript. We are thrilled to get your favorable feedback and appreciate your acknowledgment of the high quality and innovative nature of our work. We carefully read your proposed improvements and agree that modest changes are required to improve the manuscript before it is accepted for publication. We are committed to responding to each of your comments and ensuring that the final version matches the journal's high standards. We will make the necessary adjustments as soon as possible and provide a full answer to each of your comments in the updated manuscript. We feel that these modifications will improve the overall clarity and coherence of our work.

1- It is better to separate the section related to the study area in the materials and methods section and put it before the materials and methods section.

Dear Reviewer, 

Thank you for your valuable feedback, the study area section has been separated from the materials and method section. Therefore, the study area section has been moved and put it before the materials and method section. Please see line 99 in the revised manuscript. 

2- It is better to edit the text accurately by an English speaker.

Thank you for your insightful feedback. We handled the complaint about accurate language editing by having the document thoroughly examined and revised by a native English speaker. We believe that this step has substantially improved the text's precision and clarity.

Reviewer 2

The article Assessing the Environmental Factors Affecting the Sustainability of Ayni Falaj System has been read in its entirety by this reviewer and highlighted areas indicate the locations of my following comments. It is important to note that my specialty is in multivariate GIS as it pertains to ecology, however I am very familiar with the software and process described in this paper as I use them on a regular basis. I have found its use of GIS processes appropriate, innovative, and enlightening. Please see the following comments as suggestions to improve the article's readability and specificity.

Dear Kevin Lester, 

Thank you for dedicating your time to thoroughly review our article, "Assessing the Environmental Factors Affecting the Sustainability of Ayni Falaj System." We appreciate your positive remarks regarding the use of GIS processes, finding them appropriate, innovative, and enlightening. We welcome your suggestions to enhance the article's readability and specificity. Your expertise in multivariate GIS, particularly in relation to ecology, is invaluable, and we are eager to incorporate your insights. Please proceed with your comments, and we will carefully consider and implement the suggested improvements. We sincerely appreciate your constructive feedback and look forward to further refining our manuscript based on your recommendations. 

Comments:

Lines 38 - 40: These two sentences appear redundant and could be shortened to a single

sentence.

In response to your suggestion regarding lines 38-40, we have revised the manuscript accordingly. The two sentences have been condensed into a single, more concise statement while maintaining clarity and preserving the intended information. Please line 38- 39 in the revised manuscript. 

Line 49: The situation described does not have to be specific to a country and "within the

country" could be dropped.

Thank you for your feedback. Words "within the country" have been removed. Please see the revised manuscript, line 48. 

Line 53(1): "lives" could be replaced with "way of life"

Thank you so much for your feedback. Word “lives” has been replaced with “way of life”. See line 51 in the revised manuscript. 

Line 53(2): "urbanization and" falls under the category of changes to land use and could be

dropped from the sentence.

Thank you for your insightful comment. We appreciate your suggestion to improve the sentence. After careful consideration, we agree that "urbanization and" falls under the broader category of "changes in land use" and can be omitted for conciseness. The revised sentence will now read: "Another problem that this system faces in Oman is changes in land use." We believe this modification strengthens the clarity and focus of the statement. Your feedback has been valuable, and we are grateful for your input. Please see lines 51-52 in the revised manuscript. 

Line 56: "this" may be dropped from the sentence.

Done. Please see line 55 in the revised manuscript. 

Line 58: "confronts" feels inappropriate may be replaced with "faces".

Done. Please see line 57 in the revised manuscript. 

Line 189: This sentence implies the AFS is impacting the environmental variables, not the other way around which is what I think you meant. 

Thank you so much. The sentence has been corrected in the revised manuscript. Please see line 189 in the revised manuscript. 

Line 382: "Evaporation" looks to be accidently bolded.

Done. Please see line 381 in the revised manuscript. 

Line 407: "compared" may be replaced with "in relation"

Done. Please see line 406 in the revised manuscript. 

Line 451: The word "successfully" appears out of place as it is making a suggestion to the reader's interpretation and could be dropped from the sentence.

Done. Please see line 450 in the revised manuscript. 

In summary I found this article to be a novel and powerful use of multivariate GIS analysis. I believe the findings are strong and honest. Figures including the maps are appropriate and readable. Thank you for considering my comments.

Dear Kevin Lester,

Thank you for taking the time to read and comment on our article. We truly appreciate your favorable feedback and are thrilled that you found our use of multivariate GIS analysis to be new and powerful. Your recognition of the intensity and sincerity of our results is heartening since it emphasizes the time and effort we put into performing a thorough and transparent analysis. We are also glad that you found our figures, including maps, to be relevant and legible. We prioritize clarity and accessibility in visual representations, and we are pleased to learn that our efforts have been successful. We appreciate your time and thought in assessing our work. Your good feedback validates our research efforts, and we are dedicated to maintaining the quality and integrity of our work in future ventures. Thank you for your constructive feedback and support once more.

---

## [Decision Letter · Decision Letter 1]

28 Feb 2024

Assessing the Environmental Factors Affecting the Sustainability of Ayni Falaj System

PONE-D-23-30564R1

Dear Dr. Alkindi,

We’re pleased to inform you that your manuscript has been judged scientifically suitable for publication and will be formally accepted for publication once it meets all outstanding technical requirements.

Kind regards,

Abdulkader Murad, Ph.D

Academic Editor

PLOS ONE

Reviewers' comments:

Reviewer's Responses to Questions

**Comments to the Author**

1. If the authors have adequately addressed your comments raised in a previous round of review and you feel that this manuscript is now acceptable for publication, you may indicate that here to bypass the “Comments to the Author” section, enter your conflict of interest statement in the “Confidential to Editor” section, and submit your "Accept" recommendation.

Reviewer #3: All comments have been addressed

2. Is the manuscript technically sound, and do the data support the conclusions?

Reviewer #3: (No Response)

3. Has the statistical analysis been performed appropriately and rigorously? 

Reviewer #3: Yes

4. Have the authors made all data underlying the findings in their manuscript fully available?

Reviewer #3: Yes

5. Is the manuscript presented in an intelligible fashion and written in standard English?

Reviewer #3: Yes

6. Review Comments to the Author

Reviewer #3: Dear author,

I have some optional suggestions to improve your final version of the manuscript:

Please check the caption of Figure 4, because you did not mention to map b.

The content in Figure 5 appears to be similar. Make sure to check it before sending the final edition.

You also used dried and deceased words in the text. Verify which one is correct.

I believe that the paper can be improved in terms of English writing and grammar.

Thank you
